# Amway as Neoliberal Religious Tradition

**Michael Laminack**

Iliff School of Theology, University of Denver, Denver, CO 80208, USA; michael.laminack@du.edu

**Abstract:** Why do people desire their own continued oppression under neoliberalism? This essay seeks an answer to this confounding question through analysis of the Amway organization, an American multi-level-marketing (MLM) company that rose to a multi-billion dollar value in the 1980s and 90s. My argument is that Amway serves as a prime case study for the relation between neoliberalism and religious practices—-people desire their continued oppression under neoliberalism in part because neoliberalism bears meaning at the level of culture and religion. What sets Amway apart from other MLMs, and makes Amway a prime case study for neoliberalism and religious practices, is its amalgamation of neoliberal ideology with ideas and trends from American evangelicalism, to the extent that it serves as a kind of neoliberal religious tradition. As this amalgamation demonstrates, people may defend neoliberalism with a similar fervor as defending cultural or religious traditions. The conclusion explores the possibility of a decolonial American evangelicalism, which would seek options for broadening the horizons of American evangelicalism beyond the relationship to neoliberalism and the possibility of a critical theology robust enough to thoughtfully critique neoliberalism. In pursuit of this thesis, the essay utilizes a theoretical framework guided by the contributions of scholars including Wendy Brown, Walter Benjamin, Olivier Roy, Walter Mignolo, and Carl Raschke in order to analyze Amway through the lens of contemporary political theories of neoliberalism.

**Keywords:** neoliberalism; Amway; decolonization; critical theology; fundamentalism; political theory; Wendy Brown

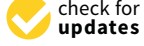

## 1. Introduction

I was raised as a religious practitioner of neoliberalism. I do not mean this in a trite way—-during my childhood, my parents joined Amway, a multi-level-marketing (MLM) business that harnessed contemporary American evangelical trends and rhetoric to recruit and motivate prospects. My parents' demographic made them prime targets for the calculated combination of neoliberal rhetoric and American evangelicalism perfected by Amway in the 1980s. In this essay, I will analyze Amway as a neoliberal religious tradition from the perspective of contemporary theories of neoliberalism. Political theorist Wendy Brown analyzes the neoliberal creation of *homo-oeconomicus*, a conception of philosophical/political subjectivity oriented purely toward economic ends. Amway serves as a prime case study for understanding how neoliberalism functions, and why neoliberal subjects ardently support and defend neoliberalism against criticism. Amway encourages members to embrace their ideology in concrete forms, through a biopolitics predicated upon consumption of Amway-produced food, vitamins, music albums, comedy albums, literature, and regular conferences. Members are further encouraged to consume Amway products while rejecting products from other companies (indeed, consuming Amway products help members reach their ultimate goal—to move up within the company by selling products). With its utilization of content from American evangelicalism, Amway functions as a religious tradition wherein the ultimate goal or outcome is the creation of a perfected *homo-oeconomicus*.

Two questions guide my thesis: Deleuze and Guattari's framing question, "why do people desire their own oppression?" and relatedly, "why, almost two decades after the MLM financially ruined my parents, do they still defend Amway?" (Guattari and

Deleuze [1972] 2009, I paraphrase the question Deleuze and Guattari ask in their analysis of Wilhelm Reich: "After centuries of exploitation, why do people still tolerate being humiliated and enslaved, to such a point, indeed, that they actually want humiliation and slavery not only for others but for themselves?"). One answer to this question, particularly with reference to defensiveness of neoliberal regimes by working-class people in the United States, is that political, business, and ecclesial leaders consciously amalgamated American evangelicalism and neoliberalism. Many working-class American evangelicals routinely defend neoliberalism with similar emotional vigor as they defend their religious tradition, because, to an extent, they view neoliberalism and their religious tradition as one and the same. Certainly, amalgamation of neoliberalism and Christianity does not occur in all instantiations of Christianity across time and space, nor even to all American evangelicals at any point in time. Nevertheless, the relationship between American evangelicalism and neoliberalism expressed in Amway sheds light on a significant undercurrent in American culture and religiosity that profoundly impacts the lives of Americans whose experiences of and commitments to American evangelicalism prevent them from thoughtfully critiquing neoliberalism and thereby welcome and defend their own oppression. Through the potent cocktail of commodified American evangelicalism, capitalist business rhetoric, and fraudulent promises of wealth creation, Amway successfully co-opted my parents, even to the point that in 2021, they still defend the organization that consumed their time and money, and led them to financial ruin.

## 2. Neoliberalism and *Homo-oeconomicus*

I grew up in a trailer park, and lack of money was an ever-present aspect of my childhood. My brother, 3 years my elder, and I avoided at all costs other students laying eyes on our run-down trailer home. When we rode the bus home from school, my brother insisted that we exit at a stop a quarter mile from our home so that other students would not see where we lived. We were also raised as KJV-only Baptist fundamentalists, a particular tradition of American evangelical fundamentalism that views the King James Version as the only authoritative translation of the Christian Bible. From a purely economic perspective, my family was perfectly primed for Amway's sales pitch. Amway promises that if members work hard enough, they will make enough passive income to retire early and spend more time with their families.[1] This vision of an ideal future lifestyle of wealth and family time also appeals to a particular vision of American evangelicalism: stay-at-home moms caring for the children while the husband works just part-time to fully provide for his family (Scheibeler 2009).[2] For working-class evangelical parents like my mother, Amway's promise appeals not just to their evangelicalism, but also to real material needs—-the desire to be home with young children instead of working full-time and scrambling to find and afford caretakers.

I will first briefly introduce Brown's analysis of neoliberalism before narrowing in on experiences within Amway as a specific case study in neoliberalism. In her 2015 book *Undoing the Demos: Neoliberalism's Stealth Revolution*, political theorist Wendy Brown presents "a theoretical consideration of the ways that neoliberalism, a peculiar form of reason that configures all aspects of existence in economic terms, is quietly undoing basic elements of democracy?" (Brown 2015, p. 17). As a political-economic theory, neoliberalism subsumes all elements of human civilization and experience under capitalist economic pursuits. By configuring "all aspects of existence in economic terms?" neoliberalism fulfills Walter Benjamin's pronouncements in "Capitalism as Religion?" which will be dealt with in a later section. Brown's analysis provides the interpretive lens for this study of Amway's amalgamation of neoliberal ideology and American evangelicalism. According to Brown, under neoliberalism "all conduct is economic conduct; all spheres of existence are framed and measured by economic terms and metrics, even when those spheres are not directly monetized?" (Brown 2015, p. 10). Thus, in Western liberal democracies, political subjects are no longer functionally *homo politicus*, but *homo oeconomicus* alone.

Neoliberal political discourses center on budgets, and civic education likewise gravitates toward economic and budgetary concerns. The political question is no longer "what is *good*?" for the *polis*/*demos*, but "what is *profitable*?" As Brown argues, neoliberalism serves to functionally rewrite Western political thought from Plato forward—-neoliberalism works to reconfigure traditional Western political philosophy by collapsing politics into economics. From a Western philosophical perspective, neoliberalism therefore reconfigures the philosophical conception of humanity. Brown writes, "In neoliberal reason and in domains governed by it, we are only and everywhere *homo oeconomicus*, which itself has a historically specific for?" (Brown 2015, p. 168). For Plato, Aristotle, and other ancients, the ideal political sphere centered on the Good. Plato's philosopher king reigns because the philosopher uniquely aligns with and seeks after the Good, and is therefore uniquely positioned to seek the Good for the *polis*. Citizenship in the Republic entails alignment with the selfsame Good (leaving room, of course, for the *others* who exist in the outskirts, and may even prove vital, but are nevertheless barred from access to the *demos*).

Opposed to these ancient Greek assumptions, "such a meaning cannot be featured in neoliberal rationality, where there are only private interests, contracts, and deals and where there is no such thing as a body politic, public good, or political culture?" (Brown 2015, p. 43). Brown builds upon this argument by analyzing Aristotle in conversation with other philosophers, including Karl Marx and Hannah Arendt, in order to show the extent to which neoliberalism reconfigures Western political philosophy. Brown argues:

As economic parameters become the only parameters for all conduct and concern, the limited form of human existence that Aristotle and later Hannah Arendt designated as "mere lif?" and that Marx called life "confined by necessity?"—concern with survival and wealth acquisition—this limited form and imaginary becomes ubiquitous and total across classes. Neoliberal rationality eliminates what these thinkers termed "the good life?" (Aristotle) or "the true realm of freedom?" (Marx), by which they did not mean luxury, leisure, or indulgence, but rather the cultivation and expression of distinctly human capacities for ethical and political freedom, creativity, unbounded reflection, or invention (Brown 2015, p. 110).

For Western liberal democracies, this theoretical pursuit of the Good helped provide rigid unity for authoritarian, religious, and monarchical societies (see: "divine right of king?"), but also provided a rallying point for radicals and revolutionaries who sought to overthrow those authoritarian societies during the eighteenth and nineteenth centuries. One's free use of reason provided access to the Good (which is necessarily rational and reasonable). By allowing free, equal access for the *demos* to pursue the Good and, ultimately, to reshape government and society to pursue the Good individually and collectively.[3]

To return to Deleuze and Guattari's question, neoliberal reconfiguration of the human subject into *homo oeconomicus* helps explain why people desire their own oppression (in this case, under a capitalist society). As Brown explains, the reconfiguration of the human subject as *homo oeconomicus* functions successfully in part because it effectively supplants the traditional political-philosophical subject with *homo oeconomicus* within a broad and generally cohesive political–economic theory. In the neoliberal schema, the human subject persists as *human capital*: "As human capital, the subject is at once in charge of itself, responsible for itself, and yet a potentially dispensable element of the whole. This is yet another way in which the social contract is turning inside out?" (Brown 2015, p. 176). Whereas the political subject of 18th- and 19th-century Western liberalism served as the bearer of universal rights and, thereby, as sovereign citizens in a democratic society, neoliberal subjects primarily serve as producers of profit. Brown argues, "neoliberalism thus does not merely privatize—turn over to the market for individual production and consumption—what was formerly publicly supported and valued. Rather, it formulates everything, everywhere, in terms of capital investment and appreciation, including and especially humans themselves?" (Brown 2015, p. 133). A "good" neoliberal subject is one who works to create financial *profit*, and profit is the only real social good. Brown refers to this reality of the neoliberal subject as "responsibilization?".

"Responsibilization?" situates the neoliberal subject into the broader neoliberal framework and motivates the subject to relentlessly pursue profitability. In this way, "Responsibilization signals a regime in which the singular human capacity for responsibility is deployed to constitute and govern subjects and through which their conduct is organized and measured, remaking and reorienting them for a neoliberal order?" (Brown 2015, p. 134). Whereas political–philosophical subjects from earlier eras of Western thought might understand themselves as citizens participating in collective pursuit of the Good (as individuals or nations/nation-states), neoliberal subjects understand themselves as individuals who must *work on themselves to become profitable*. The neoliberal subject must learn what the global market values, and fit themselves into the mold of that market demand. Brown demonstrates this aspect most clearly with respect to higher education, but anyone who has entered or even considered entering undergraduate education in the United States has likely encountered this ideology. The question "What major should I choose?" (i.e., how should I specialize my education), depends upon the job market and what industries are currently worth entering. Of course, this approach to education often fails—-job markets change, slow, or fail, or they become oversaturated with job seekers years before the undergraduate student completes their degree. If the neoliberal subject fails to properly orient themselves to the market (even if they followed "best practice?" in higher education), then they receive blame for their failure. As Brown explains, "through this bundling of agency and blame, the individual is doubly responsibilized: it is expected to fend for itself (and blamed for its failure to thrive) and expected to act for the well-being of the economy (and blamed for its failure to thrive?" (Brown 2015, p. 134).

Brown expands on her analysis of neoliberalism and the individual political-philosophical subject by introducing the work of Michel Foucault on biopolitics and neoliberalism. Brown's analysis dovetails with Foucault's by "conceiving neoliberalism as an order of normative reason?" (Brown 2015, p. 30) that works to "facilitate economic competition and growth and to economize the social, or, as Foucault puts it, to 'regulate society by the market?" (Brown 2015, p. 62). Neoliberalism is not simply a theorization of political economy that limits government intervention in free market systems, but an "order of normative reason?" that provides a consistent and coherent socio-political philosophy—-one which threatens liberal democracy itself. In this sense, Brown builds upon Foucault's conception of "political rationalities?". Brown writes, "For Foucault, political rationalities are worldchanging, hegemonic orders of normative reason, generative of subjects, markets, states, law, jurisprudence, and their relations. Political rationalities are always historically contingent, rather than necessary or teleological; however, once ascendant, they will govern as if they are complete and true until or unless challenged by another political rationality?" (Brown 2015, p. 121). Brown therefore utilizes Foucault's understanding of political rationalities as a key aspect of her analysis of neoliberalism as a pervasive and encompassing political philosophy.

It is also important to recognize that Brown does not uncritically align with Foucault, and while she shares with him an understanding of neoliberalism as an order of normative reason, Brown differs with Foucault with respect to their criticism of that order. Brown explains that "neoliberalism for Foucault was an intellectually conceived and politically implemented 'reprogramming of liberal governmentality' that first took hold in postwar Germany and was increasingly in evidence in other parts of Europe at the time of Foucault's lectures on the subject in the late 1970?" (Brown 2015, p. 50). Partly due to Foucault's experiences in this historical context, he did not develop a thoroughgoing critique of neoliberalism, and seems to have supported the regime to some extent (Behrent 2016). In line with this thinking, Foucault traced the origins of neoliberalism to modern Western liberalism itself. To describe this facet of Foucault's thought, and introduce her critical response, Brown writes:

> In Foucault's telling, liberalism was born with a market governmentality, rather than the rights of man at its heart. However, in contrast with Marxism, this governmentality rested not on the liberal state's concern with property rights,

disavowal of class, market ideology, or capture by the interests of capital, but other things altogether: on the one hand, the market was a new site of veridiction for governing and a new way of organizing, limiting, measuring, and legitimating government. On the other hand, government acquired a new and complex relationship with freedom—it produced, organized, managed, and consumed individual freedom, all without touching the subject. This is what makes liberal governmentality coterminous with the emergence of biopolitics. (Brown 2015, p. 58)

As Brown explains, Foucault's resistance to the popular Marxism of his day led him to pursue alternative socio-political theories, which then evoked in him a more appreciative conception of capital. The fact that the origins of liberal democracy led government to produce, organize, manage, and consume individual freedom, without touching the subject, lent support to capitalist society for Foucault. According to Brown, however, Foucault failed to comprehend the pervasive nature of neoliberalism, and the extent to which neoliberalism harnesses biopolitics to reconfigure the human subject and, conterminously, the entire socio-political structure. In short, Foucault failed to comprehend how neoliberalism creates new methods of domination, even as it claims individual freedom. Under neoliberalism, "rather than each individual pursuing his or her own interest and unwittingly generating collective benefit?" (the ideal model of capitalist society, and closer to what Foucault conceived), "today, it is the project of macroeconomic growth and credit enhancement to which neoliberal individuals are tethered and with which their existence as human capital must align if they are to thrive?" (Brown 2015, p. 84). As *homo oeconomicus*, individuals are not actually free to choose and decide according to their interests, but are in fact forced to sacrifice, and are chosen by and discarded by markets and market forces.

The neoliberal shift from "what is *good*?" to "what is *profitable*?" upends the undergirding principles of Western liberal democracies, but it also affects human society at the levels of culture and religion. The relation between neoliberalism and religious practices entail striking and surprising developments, particularly with respect to commodification of religion and the rise of fundamentalisms. Theorists of globalization, including Walter Mignolo and Olivier Roy, expand the argument to show how global capitalism reshapes religious practices. In order to survive, adherents adapted religious beliefs and practices for export across worldwide locales. Globalization encourages adaptations of traditional religions to the global market and, perhaps surprisingly, exacerbates the rise of fundamentalisms (Christian, Muslim, Buddhist, etc.). As religious traditions with long historical roots and deep local ties flatten to accommodate export to foreign markets (and their disparate locations and peoples), traditional adherents react in the opposite direction. Flattening and export often results in fundamentalist reactions to commodification, as adherents reassert their identity against "impure" versions of their received tradition. If the market deems a tradition *good* (i.e., *profitable*), then they are worthy of existence—all else should go the way of past, dead traditions. Perhaps ironically, the "proof" for both political traditions was/is material gain—Weber's infamous "Protestant work ethic" proved itself through the material rewards sent by God for faithful living. For neoliberalism, the infamous "invisible hand" plays a strikingly similar role as the Puritan God. A later section evaluates this fusion of neoliberalism and Christianity as a kind of "capitalist mysticism" to describe Amway's amalgamation of American evangelicalism and neoliberalism. As the following sections will detail, this aspect of neoliberalism complicates the role of religion, primarily because the free market is valueless (in the ethical/philosophical sense), and in fact erodes values that would resist neoliberalism.

### 3. Amway as Neoliberal Religious Tradition

Rich DeVos and Jay Van Andel founded the American Way Association (Amway) in 1959. They borrowed the "direct sale" strategy from Nutrilite, a vitamin company operated by Van Andel's second cousin, where DeVos and Van Andel gained business experience before forming their own company (Mondom 2018).[4] Over the following decade, the direct

sales model developed into a multi-level-marketing structure driven by distributors. In the eighties and nineties, figures within Amway increasingly focused on the business-strategy side of the organization, wherein distributors at the lower levels were compelled to purchase and consume Amway-produced content to learn how to build their business and gain wealth. This structure soon garnered attention for its resemblances to an illegal pyramid scheme. In 1983, "an Ontario Supreme Court Chief Justice fined Amway 25 million dollars, 'the largest sum (at the time) that a Canadian court has ever levied and one of the heaviest criminal penalties ever imposed against any corporation in the world?" (Grant 1988).[5] This ruling proved little more than a blip, as Amway continued to expand into a multi-national, multi-billion-dollar company.

Amway functions as a microcosm of neoliberalism. DeVos defines his business philosophy as "compassionate capitalism", a phrase that directly parallels Thatcher's ideal of capitalism with a "human face" ("What is Thatcherism?" BBC. https://www.bbc.com/news/uk-politics-22079683, accessed on 27 August 2021). "Compassionate capitalism" theoretically engages capitalism *critically* in some fascinating ways. As Davor Mondom explains, DeVos thinks "'profit' that demeans and dehumanizes our brothers and sisters or depletes and destroys the earth will lead eventually to the death of us all" and, ultimately, "all economic activity should have the goal of promoting human flourishing and self-actualization"to (Mondom 2018, p. 351). For Amway, as for neoliberalism, this promise of human flourishing and self-actualization is little more than a bait-and-switch scheme wherein members on the lower rungs work incessantly to mold themselves into the ideal compassionate-profit generators for people on the higher rungs. In theory, ideal salespeople (those who work the hardest and sacrifice the most) reach the highest wrung and achieve the utopic vision of passive income and unlimited free time. In practice, people in the lower rungs create massive distribution organizations that produce profit for their upline[6] (those on the higher rungs), while reaping little to none of the profit (Scheibeler 2009). Before returning to Deleuze and Guattari's ringing question, "Why?"—why do people, and why did/do my parents, participate, even when it clearly caused financial hardship—it is helpful to first understand *how* the organization works.

To answer the question, "How?" this section will first address how the company itself functions before addressing how Amway amalgamates capitalism and American evangelicalism to lure in potential members and then indoctrinate them into long-term company loyalty. Scheibeler explains the initial sales pitch:

> The specific example normally used to illustrate this was that of the four percent bonus. Specifically, when you help someone in your organization do 7500 PV, they 'go direct' and are getting paid 25% on the bonus scale. You then receive a leadership bonus of four percent of that distributor's personal group volume for having helped build it. Since you, yourself, are probably also doing 7500 PV, you are also receiving your own 25% bonus on all goods and services that go through your business. Your sponsor is also being paid the additional four percent leadership bonus as well. Again, it was clear that no one made more money off your efforts than you. You would receive a 25% bonus from the goods and services that went through your business, and your sponsor would receive 4% for assisting. (Scheibeler 2009, p. 22)

"PV" or "point value" refers to the amount of product the distributors move, and the goal of each distributor is to bring in more distributors to increase total PV. As more distributors enter one's "downline" (the distributors below any given distributor), the volume of product sales increases for the "upline". As a distributor moves more volume, they reach different Amway rankings: "direct distributor, silver direct distributor, gold, platinum, ruby, pearl, emerald, diamond, executive diamond, double diamond, triple diamond, up to the highest level, a crown ambassador" (Palmisano and Pannofino 2013, p. 31). Distributors who move up the rankings are promised a yearly income of USD 25,000 and up, on part-time work (12–15 h per week) (Scheibeler 2009).[7]

Once distributors reach the Direct Distributor status, members attend a special seminar where they learn the "tool" of the business (Butterfield 1985, p. 16). These tools consist of books, cassette tapes, and other media produced by business people and motivational speakers, all of which ostensibly teach the distributor how to define and achieve their financial dream.[8] The tools help garner company-loyalty in part by indoctrinating members with the criticism of capitalism proffered by DeVos's compassionate capitalism ideology. These Amway-sponsored materials denigrate common workers of all stripes who are merely "exchanging time for dollars" while building equity for their bosses and none for themselves (Scheibeler 2009).[9] Distributors watch, read, and listen to these tools, all of which they purchase through their upline, for hundreds of hours. Their upline likewise impels them to attend monthly seminars and rallies, where distributors receive further instruction. Higher levels (particularly Emeralds and Diamonds) flaunt their Amway-gained wealth at these rallies, and encourage distributors to work as hard as necessary to achieve their great life of passive income and unlimited family rest time. As most ex-Amway members have shown, however, "the vast majority of distributors donate a great deal of free labor to the Company and make nothing" (Butterfield 1985, pp. 18–19).

Beyond the basic structure of the company, Amway tools function as a fine-tuned example of Foucault's biopolitics. The tools compel distributors in their daily actions (*constant* work to recruit new members), but the tools also compel distributors to transform their physical appearance to fit the Amway mold. Scheibeler explains that "no leaders were permitted to have facial hair" (Scheibeler 2009, p. 98). Likewise, Butterfield states: "committed distributors are ones who 'duplicate properly,' that is, do precisely what they are told. They cut their hair the way they are told to cut their hair. They spend a minimum of three nights a week recruiting, a day for product flow and a day for retail sales. They follow directions exactly as give" (Butterfield 1985, p. 31). This aspect particularly strikes a chord with me, as my first memory of anything related to Amway is seeing my father after he returned from his first seminar. When he walked in the front door of our trailer, I saw his clean-shaven face and I promptly turned and ran back to my room. This may seem an unimportant detail, but this speaks to the degree of minute biopolitics involved in Amway. To reach the Dream, you must fully commit—down to the minutest of personal and physical details.

## 4. Amway and Capitalist Mysticism

The manner in which Amway amalgamates American evangelicalism and neoliberalism deserves scrutiny as a phenomenon which I will refer to as "capitalist mysticism".[10] This mysticism involves three main aspects: prosperity theology, saintly knowledge, and mystic discipline. With respect to prosperity theology, Colin Grant refers to how Amway grounds its ethical emphasis "in a variation on the late Old Testament obedience—prosperity/disobedience—disaster theology" wherein the success of Amway is attributable ... ultimately to obedience to the divine design of life itself" (Grant 1988, p. 490, Grant quotes from Ian Austen. Looking into Amway's Empire. Macleans, September 6, 1982; p. 28). Amway's prosperity theology differs from other related forms of Christian prosperity-centered theologies because Amway does not call practitioners to pray or read Scripture, but to follow the "tool?" and sell the business. The gnostic element of saintly knowledge also refers to this process of "duplication"—-according to Amway teaching in the tools, jewel-level distributors, especially Diamonds, have gained and practiced the necessary knowledge of the business, and have thus attained their dream. Butterfield explicitly draws the connection to evangelical religious practices by describing how at seminars and rallies, "the Personal Story is seen to be a definite *genre*, a confessional autobiography, similar in purpose to the testimony of born-again Christians on PTL television program" (Butterfield 1985, p. 40). Distributors listen to these stories and emulate the saints who tell them in order to emulate them, and, hopefully, duplicate their success. In various traditions of Christianity, saintly figures deserve emulation because they lived in a manner that brought them closer to God. For Amway saints the same logic applies, except their

sainthood is proven not by otherworldly sacrifice or biblical wisdom, but by the wealth and status attained through the business.

Amway literalizes what Walter Benjamin described in his posthumously published 1921 essay "Capitalism as Religion". In this essay, Benjamin argues against Max Weber's description of capitalism "as a formation conditioned by religion" as with Weber's oft-quoted "Protestant work ethic" (Benjamin 2004, p. 288). Benjamin instead approaches capitalism "as an essentially religious phenomenon" (ibid., p. 288) unlike Weber's notion of religion as an accidental relation to capitalism's rise (Benjamin 2004). Benjamin continues on to describe "three aspects of this religious structure of capitalism":

> In the first place capitalism is a purely cultic religion, perhaps the most extreme that ever existed. In capitalism, things have a meaning only in their relationship to the cult; capitalism has no specific body of dogma, no theology. It is from this point of view that utilitarianism acquires its religious overtones. This concretization of cult is connected with a second feature of capitalism: the permanence of the cult. Capitalism is the celebration of a cult *sans rêve et sans merci* (without dream or mercy). There are no 'weekdays.' There is no day that is not a feast day, in the terrible sense that all its sacred pomp is unfolded before us; each day commands the utter fealty of each worshiper. And third, the cult makes guilt pervasive. Capitalism is probably the first instance of a cult that creates guilt, not atonement. (Benjamin 2004, p. 288)

Each of the aspects described by Benjamin here are deepened and amplified by later theorists of neoliberalism. All three aspects imply, in their disparate ways, the suffocating presence and all-encompassing nature of capitalism. Capitalism has no dogma or theology, which means that every possible dogma or theology can be taken on (commodified) by capitalism. There are likewise no holy-days, no rest, and no cycle, as with most if not all faith traditions. Ultimately, Benjamin highlights the type of mystical thinking about capitalism that reaches its apotheosis with Amway.

Benjamin's analysis also dovetails with Olivier Roy's analysis in his *Holy Ignorance: When Religion and Culture Part Ways*. Benjamin develops his description of capitalism as religion by further contesting Weber's conception of the relationship between Christianity and capitalism: "Capitalism has developed as a parasite of Christianity in the West (this must be shown not just in the case of Calvinism, but in the other orthodox Christian churches), until it reached the point where Christianity's history is essentially that of its parasite—-that is to say, of capitalism" (Benjamin 2004, p. 289). For Benjamin, Protestant faith may have practically benefited the rise of capitalism, but capitalism then devoured that faith through an amalgamation of capitalist political economy and Protestant theology and practices. Roy describes a similar shift in the era of neoliberal globalization, wherein the alteration of Protestant Christian faith and practice described by Benjamin now occurs at a global scale and with global consequences. Roy's more thorough analysis demonstrates that Benjamin's brief reflection overstates the relationship between Protestant faith and capitalism, as numerous neoliberalism-resistant indigenous and decolonial theologies (Protestant and otherwise) attest. Yet, alterations of traditional faiths in various global contexts, including but not limited to Christianity, likewise demonstrate the profound impact of neoliberalism on traditional faiths and cultures. One obvious consequence is the rise of various religio-political fundamentalisms. Roy reasons that "fundamentalism is the religious form that is most suited to globalization, because it accepts its own deculturation and makes it the instrument of its claim to universality" (Roy 2014, p. 5).

One of the key terms Roy introduces to describe the relation between religion and culture in the era of globalization is "deculturation" which names the transition from traditional religion to fundamentalism. Roy helpfully defines his understanding of "culture" as "the productions of symbolic systems, imaginative representations and institutions specific to a society" and, he adds, "religion is treated by anthropologists and sociologists as one of several symbolic systems; it is therefore seen as an integral part of a given culture;

it is of the culture" ([Roy 2014](), p. 26). In addition to this basic understanding, there are multiple ways in which religion interacts with the culture. Roy explains,

> Each time there has been a questioning of the relations between religion and culture, prefixes have been added to the word 'culture': to *de*culturate, *ac*culturate, *in*culturate, and *ex*culturate. Religion deculturates when it attempts to eradicate paganism (conquering Christianity in America, orthodox Islam on the Indian subcontinent); it acculturates when it adapts to the mainstream culture (the Jews of the *Haskala* (Enlightenment), Christianity and Islam in India); it inculturates when it tries to establish itself at the centre of a given culture (the theologians of Latin America's 'indigenous' Christianity), and it exculturates when it thinks of itself as standing back from a mainstream culture of which it was part, but which suddenly or gradually took on a negative, 'pagan' or irreligious—and therefore destructive—aspect (Catholic and evangelical reaction at he close of the 20th century, the Tablighi Jamaat movement within Islam). ([Roy 2014](), p. 33)

Roy later pairs the concept of deculturation with deterritorialization (the excision of a religion from its place in addition to culture), both of which occur in a manner that helps fundamentalisms thrive in the globalized era.

What makes Roy's analysis crucial for understanding the appeal of Amway to American evangelicals (and/or fundamentalists) is the tension between the culture and religion, and how deculturation weakens the knowledge of adherents and opens them to commodified forms of their faith. Roy describes this phenomenon as a "weakening of the horizon of intelligibility" wherein "the ties between religion and culture are severed: in the eyes of religion, culture ceases to be profane and becomes pagan" and "is both a consequence and an instrument of globalization and it largely explains the success of fundamentalist forms of religion" ([Roy 2014](), p. 115). The move from traditional faiths to fundamentalism entails a move from viewing the relationship between religion and culture through the lens of sacred/profane to sacred/*pagan*, where the latter suggests an existential threat to the faith. United States history offers numerous historical examples of these processes, as various Christian communities adapted to economic and social changes and often in response to significant waves of immigration. In these moments, American evangelical leaders argued that culture had degraded, and churches underwent revivals in response, and these revivals eventually birthed modern American evangelical fundamentalism.

One of the surprising implications of Roy's argument is that there persists a close relationship between globalization and the rise of fundamentalisms. Indeed, it seems that fundamentalism responds to globalization, but, at the same time, fundamentalism also *primes* adherents for commodified versions of their faith. Roy comments on this phenomenon by asserting that "there is a paradox: those who return to religion, as converts or as born-agains, do so without religious knowledge, which they may or may not subsequently acquire, but it will be a knowledge divorced from any cultural context. The erosion of religious knowledge in fundamentalist circles is particularly striking" ([Roy 2014](), p. 120). This description encapsulates what drew my parents to Amway, though with some minor alterations—my parents, raised in Baptist fundamentalism, lacked the knowledge and necessary tools to recognize how Amway commodified Christianity. For American evangelical fundamentalism, criticism of culture has definite limits. American patriotism, specifically, blurs the fundamentalist boundaries not between profane or pagan, but profane or sacred (notice, for instance, the ubiquity of American flags adorning sanctuaries of American evangelical fundamentalist churches). Roy accounts for this phenomenon by, like Benjamin, referencing Weber: "Ultimately, in defining a 'Protestant ethic', instead of producing a work of history, Weber ended up formatting Protestantism for exportation within the framework of capitalist globalization" ([Roy 2014](), p. 197). Partly due to the privileged place of Christianity in American society, as well as the historical importance of the faith, fundamentalist criticisms of American culture will differ from, say, criticisms in/from Islamic fundamentalisms. In the American context, Christian fundamentalism

does not criticize, but actively *protects* aspects of American culture. This protection includes, to a significant degree, the economic system of capitalism.

Benjamin briefly addresses this interplay between certain Christian traditions and capitalism:

> The Christianity of the Reformation period did not favor the growth of capitalism; instead it transformed itself into capitalism. Methodologically, one should begin by investigating the links between myth and money throughout the course of history, to the point where money had drawn so many elements from Christianity that it could establish its own myth. (Benjamin 2004, p. 290)

Reformation-era Christianity indeed did not favor the growth of capitalism, but in fact threatened its growth—-in addition to generally threatening the European power structures that enabled capital accumulation, these new theologies helped spur particular events like the German Peasants Revolt as well as the various wars of 1848. But certain forms Reformation-era Christianity did conform themselves to capitalism, and this is made abundantly clear through the protectionist relation between American evangelical fundamentalism and capitalism. American evangelicals such as my parents may experience no cognitive dissonance when presented with the argument that free markets determine the value of people and their labor, though their confessed theology would grant that determining power to God alone. Thus, the "invisible hand" of the marketplace and the God of the Christian Bible are difficult to disentangle, and American evangelical fundamentalism provided few conceptual tools to help my parents criticize, question, or even recognize such entanglement.

American evangelical fundamentalism complicates the sacred/profane sacred/pagan categories that Oliver Roy describes. At least since the Bolshevik Revolution and the subsequent First Red Scare, American evangelical fundamentalists sought to defend the sacred American ideal from the taint of pagan, godless Communism. Amway showcases this theo-political entwinement—-at seminars and rallies, distributors are "bold in proclaiming their faith and patriotism" (Scheibeler 2009, p. 4) and "people open every seminar with a prayer and a pledge to the flag" (Scheibeler 2009, p. 89) and each one ends with participants holding hands and singing "God Bless America" (Scheibeler 2009; Butterfield 1985, p. 42). In addition to this entwinement of faith and patriotism, which is a common aspect of American evangelicalism, Amway proffers explicit support for DeVos's neoliberal-tinged variety of capitalism. Since Amway couches its system in the language of compassionate capitalism, distributors also view their actions as "helping others succeed" in reaching their dreams and becoming wealthy (Scheibeler 2009, p. 108). Distributors seek personal wealth and success, but Amway allows them to do so *altruistically*—everyone who enters Amway will also receive the opportunity to live the blessed life of Amway.

Stefania Palmisano and Nicola Pannofino, researchers who analyze Amway in Italy, describe Amway as a "quasi-religious corporation" a phrase that refers to organizations that make "common reference to a cultural system that has Protestant religious roots" and which hinges "on the sense of a divine mission entrusted to the American people, and thus the propagation beyond national borders of the 'American way of life'" (Palmisano and Pannofino 2013, pp. 27–28). Grounded in this "mission" Amway has a "double economic–expressive nature" (Palmisano and Pannofino 2013, p. 35), a phrase that describes how Amway combines economic incentives with the work on the self for distributors in the organization (Palmisano and Pannofino 2013). What sets Amway apart from other multi-level marketing schemes is "the emphasis which it places on these characteristics by circulating books and videos presenting their sales and distribution efforts as a real lifestyle requiring members' involvement and dedication" (Palmisano and Pannofino 2013, p. 31). For American evangelicals, this model closely resembles their faith tradition, and Amway utilizes traditions and trends in American evangelicalism in service of its business. Unlike evangelicalism, wherein one works toward self-transformation through regular prayer, scripture-reading, and church attendance, for Amway practitioners "personal development goes hand-in-hand with economic success" (Palmisano and Pannofino 2013, p. 42).

Palmisano and Pannofino further describe Amway as an "identity transformation organization" which signifies "an organization which requires, as a condition of belonging, a change in self-ideation. Adherence and a sense of belonging, in their turn, are mediated by identification processes with which the organization shows its favour". (Palmisano and Pannofino 2013, p. 37.) For this identification process in Amway, a distributor must: "1. Read at least 15 min a day from a book on the 'tool list' 2. Listen to a tape everyday 3. Attend all functions (seminars, training sessions etc.) 4. 100 percent self-use of products in our home 5. Show the plan to people we would like to help" (Scheibeler 2009, p. 27). This identification process mirrors what an American evangelical pastor implores a congregation to do: 1. Read the Bible or devotional materials daily 2. Listen to worshipful music or devotional content 3. Attend Sunday church services and other weekly activities 4. Practice the faith throughout the week, including at work 5. Proselytize family, friends, and strangers. In a 1988 article, Colin Grant described Amway's emphasis on the "ethical" in the business as "a variation on the late Old Testament obedience–prosperity/disobedience–disaster theology" whereby "the success of Amway is attributable...ultimately to obedience to the divine design of life itself" (Grant 1988, p. 490). Amway effectively amalgamates this model of American evangelicalism and capitalism by redirecting evangelism toward the production of profit rather than the saving of souls or serving the Kingdom of God.

The structure of seminars and rallies also mimic trends in American evangelicalism. Palmisano and Pannofino rightly state that "for Amway members, taking part in the meetings is a ritual and ceremonial event of central importance" (Palmisano and Pannofino 2013, p. 40). As previously mentioned, speakers bookend the meetings with explicit references to American evangelicalism (beginning with prayer and recitation of the pledge, ending with "God Bless America" along with plenty of references peppered throughout). Attendees describe feeling "a euphoric mixture of motivation and exhaustion" partly from the fact that attendees are obligated to attend meetings past midnight. Some of the seminars also included Sunday morning "church" services. These services included an "altar call" an evangelical tradition dating back to Charles Finney's "anxious bench" from the 19th century Second Great Awakening. At the end of a service, the leader invites attendees to approach the altar at the front of the sanctuary and commit or recommit their lives to Christ. Scheibeler describes his experience during one such altar call:

> At a large Dream Weekend seminar for Amway distributors, there was an emotional Sunday service that ended with an altar call. Hundreds began to move forward. I wanted to go up, but at the same time, I felt paralyzed by fear. At the last minute, I left my seat and went forward and recommitted my life to Christ. I was working so hard but something was wrong, because I seemed to be the only person not making a strong income to help my family. From the indoctrination, I believed that I had a spiritual problem: a lack of true faith that was blocking me from being the husband, father and provider that I was called to be. This was to be the first of several trips I made to the altar over the next few years. Each experience was more emotional than the last. I felt completely drained, not knowing why I was not succeeding. I was desperate to find the solution. Surely, God would hear my prayers. I was spending far more time serving His people than with my own family. (Scheibeler 2009, p. 74)

While Amway overtly utilized these aspects of American evangelicalism to consciously appeal to people like my parents, Amway also establishes an us-versus-them rhetoric that likewise resembles American evangelical fundamentalism. Distributors are taught to purchase *only* Amway products (partly to increase sales volume, and partly to set a good "example" for their downline), and to avoid "negative"[11] (non-Amway) products (Scheibeler 2009). In some corners of the organization, Amway members are encouraged to buy and imbibe Amway-produced media, including music and even comedy specials. Butterfield mentions hearing music from "the Sammy Hall Singers" Butterfield explains, "Sammy Hall is a born-again ex-hippie who writes songs specifically for Amway functions (Butterfield 1985, p. 25)". On any given trip, my parents would play the music of

Dreamer, a Plano, Texas group who wrote and produced Amway-related music along with uplifting evangelical hymns. The Amway-related songs included parodies of popular music, including "Friends Going Places", an adaptation of Garth Brooks's "Friends in Low Places". Once more, this music mimicked a popular evangelical trend of producing evangelical-friendly versions of songs or music styles—-Christian companies send youth pastors posters of popular musicians and their "Christian alternative" deemed acceptable for evangelical youth.

The us-versus-them mentality around products and media also extends to other people and to politics. For distributors, people inside and people outside the business occupy "two separate and distinct worlds" (Scheibeler 2009, p. 58). Every person outside the business becomes a "prospect" and, if the distributor embraces the Amway lifestyle, "there are only four categories of people: prospects, distributors, customers and losers" (Butterfield 1985, p. 64). Those who refuse any involvement with the business are "losers" for rejecting the Amway vision, and are therefore choosing to be poor. At seminars and rallies, and in the "tools" distributors receive "education about politics and unions" including encouragement to "vote correctly"—meaning to vote for and "contribute financially to the campaigns of those conservative Republicans who were brought in to speak to us" whose ultimate goal was "to get the whole country right" (Scheibeler 2009, p. 43). Both the view of "outsider" and this approach to politics appealed to my parents' fundamentalist background—fundamentalist sermons routinely reference the nefarious "them": "they" reject God, "they" want to corrupt your kids and destroy your culture and country. The anti-Communist rhetoric morphs into general anti-socialist rhetoric that eventually incorporates any pro-union or pro-worker policies or positions.

Interestingly, Butterfield's reflections on Amway forecast Brown's analysis of neoliberalism in some fascinating ways. In a particularly prescient statement, Butterfield writes, "Amway must mark some kind of watershed in the history of applied political thought. To glimpse what a Reaganite future would mean for working people, perhaps we can regard the Amway plastic bubble as a crystal ball" (Butterfield 1985, p. 138). This neoliberal future would entail a large consumer/working-class who must work (*for* the business and *on* the self) incessantly to reach the upper rungs of society, rungs available only to the faithful few willing to sacrifice everything to receive the prosperity promised by the god of capital. Reflecting on these possibilities, Butterfield interrogates the implications of a neoliberal regime and issues a warning that resembles Brown's arguments regarding neoliberalism and liberal democracy:

> But what is lost when I root out a wage earner's values from the self and replace them with corporate management thinking and belief? What values am I giving up? Is there something vital and positive about the attitudes on the lower end of the social scale that workers need in order to safeguard their own interests as workers? What does it mean in social terms for the Amway ethos to be vigorously, persistently, and with missionary zeal, propagated throughout a working and middle class population? What long-range effects will this have on support for public schooling, the strength of organized labor to resist oppression and paternalism from big business, the level of wages and benefits for all employees, the preservation of civil liberties, national resource land and social services? The time to answer these questions is now, before Ronald Reagan appears the most liberal of the candidates to hold office in the United States of Amway. (Butterfield 1985, pp. 131–32)

## 5. The Possibility of Decolonial American Evangelicalism

Is there hope for American evangelicalism to critically engage neoliberalism, when neoliberalism commodifies even extreme forms of fundamentalist adherents to support their own oppression? In *The Darker Side of Western Modernity*, Walter Mignolo utilizes Aníbal Quijano's analysis of the "patrón colonial de poder" or "colonial matrix of power" that constitutes modern Western civilization, with "four interrelated domains: control of



the economy, of authority, of gender and sexuality, and of knowledge and subjectivity" (Mignolo 2011, p. 8). To attempt to challenge and change the global hegemony of neoliberalism, Mignolo calls for "moving toward decolonial horizon" (Mignolo 2011, p. 33.) that confront each domain of the colonial matrix (Mignolo 2011). This type of action that Mignolo advocates are also "pluriversal, not universal" which "take pluriversality as a universal project to which all contending options would have to accept" as they "think and act decolonially" (Mignolo 2011, pp. 23–24). Is it possible for American evangelicalism to take part in this neoliberalism-resistant, pluralist landscape? Decolonial scholars and activists may deny such a possibility—-how could American evangelicalism play a role in dismantling the colonial matrix it helped found and perpetuate? This is a profoundly difficult question, given the history and contemporary circumstances of American evangelical support for neoliberal policies and ideas, and Christian scholars should not dismiss such questions without serious consideration and reflection.

As this study has shown, American evangelicalism continues to provide crucial support for neoliberalism—-for practitioners of Amway, as well as American practitioners of neoliberalism less committed to evangelicalism or any other form of faith, Christianity and neoliberalism are difficult to dissociate. To attack neoliberalism is to attack Christianity, or to attack their conception of American culture itself, and this is further compounded when scholars describe this amalgamation from an outside position. Practitioners will defend neoliberalism with the same ferocity as they would defend their religion, country, or family. The irony of this reality, of course, is that neoliberalism adapts Christianity as it commodifies it. Fundamentalists who decry the destruction of "Christian value" by the Left fail to see the destruction of those values by the neoliberal regimes they rabidly support. American evangelical fundamentalists criticize society, which often earns them social derision, but with respect to decolonial thinking, they are *not critical enough*. Raised in Christian evangelical fundamentalism, my parents lacked tools for either theological/ethical criticism or economic criticism and, oddly, fundamentalism helped make my parents *more* susceptible to Amway.

Is a decolonial American evangelicalism *possible*? Carl Raschke beckons toward such a possibility with his call for "critical theology" (Raschke 2016). As Raschke explains, a critical theology "has to detect and articulate the nature of the *force* behind faith" and, in so doing, "a critical theology is *emancipatory* in the measure that it frees us from all the ideological conceits and theoretical modes of *overdetermining* the human condition that so many secular and influential non-theological 'theories' routinely make" (Raschke 2016, pp. 74, 110). Critical theology opens the possibility of American evangelical engagement in the pluriverse of thinkers and activists working toward decolonial horizons. Through this critical engagement, American evangelicals may find that capitalism is the "hegemonic structure of the past five hundred years; this doesn't mean that it is the only option or the best" (Mignolo 2011, p. 304). While this hegemonic structure currently commodifies much of American evangelicalism, neoliberalism, and American evangelicalism are *not* one and the same. The broader fabric of Christian tradition provides a view of human purpose that is not necessarily to be a "consumer-subject" who "live to work and work to consume, instead of working and consuming to live" (Mignolo 2011, p. 36). American evangelicalism has not always taught this, and it does not have to in the future. Pursuit of critical theology directed toward decolonial horizons offers hope for an American evangelical future in the pluriverse of decolonial options. While many Christian leaders and adherents will undoubtedly resist such pursuits, those leaders and adherents, including my parents, would only benefit from a future wherein their faith served their emancipation, and not their oppression.

**Funding:** This research received no external funding.

**Institutional Review Board Statement:** Not applicable.

**Informed Consent Statement:** Not applicable.

**Data Availability Statement:** Not applicable.

**Conflicts of Interest:** The author declares no conflict of interest.

## Notes

1　(Scheibeler 2009, pp. 9, 59). While I will present information from personal experiences, I will also utilize publications that present firsthand accounts of Amway. I cannot vouch for everything these authors claim in their full works, but I will reference parts of their works that corroborate my experiences and the recollections of my parents.

2　(Scheibeler 2009, pp. 17). Scheibeler describes how the distributor (the pseudonymous Kerry) who hooked him and his wife by appealing to this vision: "This venture was looking like a dream come true. Kerry and Chris loaned us a copy of Profiles of Success to keep until we received the one that we had ordered. We stayed awake late every night reading each Diamond's success story. Most all had struggled before they entered The Business. From the book and the tapes that we were now avidly listening to, we learned that these people had become full-time parents. Making money was not the highest priority in their lives. Most all spoke about "the system?". Many praised God for their new lives. I was not a deeply spiritual person at that point, but it gave me a sense of reassurance that we were dealing with people of faith and integrity?" (17).

3　Again, it must be noted that the ancients bequeathed a tradition wherein others are included in the scheme, but are excluded from the demos (and therefore from political decision). Thus, when the language of creator-endowed "natural right?" problematized traditional exclusionary conventions, theoreticians sought new theories and methods to justify the continuation of traditional exclusions. New sciences determined that non-White, non-males naturally lacked access to reason and, thereby, lacked the necessary qualities of humans who can enter into social contracts.

4　(Mondom 2018, p. 348). Mondom's article offers an excellent history of Amway from its inception to its current impact on American politics.

5　(Grant 1988, p. 490). Grant quotes from Ian Austen, 'Looking into Amway's Empire', *Macleans*, September 6, 1982, p. 28.

6　(Scheibeler 2009, p. 50). "The next rule requires a basic understanding of the network or multilevel marketing structure. As mentioned before, your sponsors and the people above them were your upline and had a vested interest in your success. The distributors that you sponsored and those below them were your downline and were distributorships in which you had a vested interest. Any other distributor would be considered cross line. For example, let's say your sponsor sponsored you and another couple named Bob and Mary. Bob and Mary, and the entire organization that they developed, would be considered cross line to you. "Never cross line?" was a core principle that referred to not having business-related or personal conversations with distributors that were cross line from you?" (emphasis removed).

7　(Scheibeler 2009, p. 59). It is worth noting that Scheibeler entered Amway from a comfortable middle-class position. There are understated class requirements for MLMs such as Amway that preclude people like my parents from ever attaining "success?" (even the mirage of success that Amway promises). To sell Amway to potential distributors, members must visually demonstrate their own success. Successful Amway salespeople benefit from middle– or upper-middle-class homes to which they can invite others to deliver their pitch. My parents were never able to sell this image—no one was coming to an old trailer to learn about financial success.

8　(Mondom 2018, pp. 350–51). "The company's corporate philosophy found expression in books authored by the cofounders and 'Amway sympathizers'—a term fashioned by Charles Conn to refer to those not directly affiliated with the corporation but who championed it—as well as audiocassette tapes of addresses given at Amway functions, typically by distributor couples but also by motivational speakers. 31 Many of these items fall under the rubric of what, in Amway circles, are called [351] "tool?". Tools, according to Shad Helmstetter, a pro-Amway motivational speaker, are an 'essential' component of an Amway distributorship, something to consume daily?".

9　Scheibeler writes, "Zack [a pseudonym for Scheibeler's Diamond-level upline] went on to explain many people's lack of success is due to our engaging in production work of one sort or another. Specifically, most of us toiled our entire lives, exchanging time for dollars. It would not matter if we were bricklayers or neurosurgeons, as both merely traded hours for dollars. No matter how long we did this, we would always be busy. Zack used many analogies to illustrate the point that most people were working harder and longer for less and less. He emphasized that, according to the Social Security Administration, 95% of the people at retirement would be either dead, dead broke, or still working. Only five percent would ever become financially free. How could anyone succeed against such staggering odds?", (Scheibeler 2009, p. 13).

10　(Darlington 2016; Torre 2014). Darlington uses the phrase in reference to science fiction, a context which seems unrelated to my argument. Torre uses the phrase in reference to Adam Smith and the idea of the "invisible hand?" in free market systems. The meaning I intend here therefore aligns fairly well with Torre. It is worth noting, though, that Torre uses the phrase in a passing comment on Smith's capitalism. The phrase deserves further development, which I hope to begin here.

11　(Scheibeler 2009, p. 38). "Products purchased outside of Amway were referred to as "negative?" products. We were told that distributors would even look in the closets in our bathroom when they were over for meetings. We certainly could not afford to have them find a negative product and think that we did not believe in The Business. There were even little, orange, "hazardous material?" negative, product stickers that you could order that had a skull and crossbones across them and said something like, "this product may be hazardous to your PV?". One prominent Florida Diamond talked about having negative PV raids on the

homes of his downline distributors. It sounded like it was done in fun, but they would actually run through the distributors' houses, either labeling or collecting all non-Amway products. This was serious.

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
