# Peer review of "Amway as Neoliberal Religious Tradition"

_religions, doi:10.3390/rel12090703_

Round 1

Reviewer 1 Report

I judge this to be a very interesting, well-documented, mostly coherent (see below), and persusaive paper.  The connections between Amway, a certain kind of American evanglicalism, and neoliberalism are well made. The engagement with major figures such as Foucault and Benjamin helps to locate the arugment in a larger horizon of cultural criticism, and the engagement with more recent scholars (Brown, Roy etc) provides a freshness and immediacy to the discussion. In addition to the few very minor grammatical and style issues I have added to the manuscript, I mention the following three points.

  1. I think the title deserves to be more interrogative than its present descriptive mode. As the first sentence of the Abstract indicates, there is a question driving this paper (and it is answered) and it warrants being flagged in title. See my suggestion in my annotation.
  2. I have some concerns about the biographical elements in the paper. It is fine for the author to draw on his/her own experiences of growing up in the Amway/Evangelical culture, but he/she also  makes claims about his/her deceased grandfather (and a power struggle he was involved in, p.9 ), his father (his vulnerabilty to Amway, p. 9) and his mother (her anger p.2) all of which have borderline moral judgements attached to them. There is no acknowledgement that these details and claims are made with the father's and mother's approval. I recognize that it's borderline, but in my own academic context such information would require approaval from a Human  Research Ethics Committee. 
  3. The most significant issue for the coherence of the paper is the final and concluding section. Its heading refers to Decolonial  Christianity and begins by asking whether there is any hope for Christiainity. This is odd because it was not the presenting issue of the paper, which related to why Amway could so effectively indoctrinate people. The discussion about decolonising Christianity is not irrelevant to the paper, but it needs to be introduced in ways that make its connection to the paper's stated aim much clearer and coherent. 

Author Response

Response to Reviewer 1

I am grateful for the comments of the reviewers––both the positive statements and the incisive, insightful criticisms. Thank you for helping to improve this manuscript. I have broken down the critical comments into six points below. If I have somehow missed something, it was unintentional and I am happy to address further issues. 

I found the introduction of the 'decolonial' frame in the conclusion a little disconcerting. I had not considered the paper as a contribution to scholarship on decolonizing religion. Neither the abstract nor the introduction had indicated this. As a consequence, the introduction of 'decolonial Christianity' in the conclusion seemed out of place. This can be easily fixed by foregrounding this dynamic much earlier in the paper, and clearly signposting the conclusion as the culmination of the argument. 

  1. Introduce decolonial Christianity earlier and in abstract (Both reviewers)

This is a clear deficiency that I will fix by adding a few sentences peppered throughout the paper and in the abstract. 

Two further features about the conclusion also caused me concern. The first was the affirmation that Christianity cannot play a role in "dismantling the colonial matrix it helped found". This is framed as "a fair perspective" that ought to be accepted as valid. But the author clearly disagrees with this view, argument just a few paragraphs later that "neoliberalism and Christianity are not one and the same" and that a decoupling of Christianity and neoliberalism is a necessary measure. This can't be held both ways; and a clearer concluding argument is important.

  1. Deal with “fair argument” against decolonial Christianity (Both reviewers)

The ambiguity here arises from the criticism of Christianity by non-Christian and post-Christian indigenous scholars (particularly George “Tink” Tinker), as well as my own, albeit unstated, philosophical stance. Ultimately, I see the possibility of decolonial Christianity (particularly from American evangelicals) as a groping in the darkness––I am hopeful, and eager to put forth the effort in its pursuit, but it is an enormous and unending task. It’s “accomplishment” entails a permanent process of revolution and rethinking in the face of new circumstances and criticisms, and it can never erase the history of atrocities committed. Nonetheless, I recognize that this kind of paradoxical approach appears merely contradictory in an article like this. I will address the issue as briefly as possible (i.e., without the explanation above) in the manuscript. 

 Two further features about the conclusion also caused me concern. The first was the affirmation that Christianity cannot play a role in "dismantling the colonial matrix it helped found". This is framed as "a fair perspective" that ought to be accepted as valid. But the author clearly disagrees with this view, argument just a few paragraphs later that "neoliberalism and Christianity are not one and the same" and that a decoupling of Christianity and neoliberalism is a necessary measure. This can't be held both ways; and a clearer concluding argument is important.

Further, the argument that Christianity is irredeemably neoliberal/colonial is clearly an attempt to delegitimize and silence all those voices associated with Christianity. This can only be seen as a colonizing secular manoeuvre to coercively impose hegemonic discursive limits to political participation. In so doing it erases, for example, most of the indigenous voices of the Pacific who live and engage in the politics of decolonial liberation via the paradox of Christianity as both colonizing and decolonizing. As a mode of modern purification, the silencing of indigenous Christian voices advocated here is ethically dubious at best.     

It should be possible to pose a challenge about the possibility of decolonial Christianity without overstating this.

Second, also in the conclusion, while the coupling of Christianity and neoliberalism in the US context has been extensive, it is not helpful to argue that "perhaps" for neoliberalists "Christianity and neoliberalism are one and the same." This again overstates the argument, by rendering the American context into a universal one. A qualification here about the American locations of this statement would be valuable. 

For example, it is clearly not the case that Buddhist practitioners at the Dhammakaya Temple in Thailand, as described by Rachelle Scott in her book Nirvana for Sale? Buddhism, Wealth and the Dhammakaya Temple in Contemporary Thailand (2009) imagine themselves as Christian. Nor is this the case for Muslim activists in Egypt as described by Mona Atia in her book Building a House in Heaven: Pious Neoliberalism and Islamic Charity in Egypt (2013). Of course, the striking continuities between the author's argument in this paper and those described by Scott and especially Atia are thoroughly revealing for the ways in which neoliberal logics have become insinuated within religious dynamics across a diverse array of traditions. 

  1. Silencing of indigenous Christianity; universalizing American experience (Reviewer 1, first poin)

I am extremely grateful for this critique. I did not intend to universalize the American experience, and I in no way seek to silence and/or delegitimize indigenous forms of Christianity. I failed to specify that I intend to address White, American evangelicalism (which is certainly still overdetermined and generalized, but also too narrow at the same time––White, American evangelicalism undoubtedly shares qualities and histories with non-White American Christian traditions. The long history of televangelists and “prosperity gospel” traditions are by no means limited to White, American evangelicalism – and are certainly at play with an analysis of Amway and neoliberalism – but that analysis is beyond my capacity as a scholar and a person.  I will add specificity, hopefully without too heavy a hand, to address this excellent critique.  

Finally, as I read the article I was frequently struck by similarities to the argument detailed in Beth Morton's book To Serve God and Wal-Mart: The Making of Christian Free Enterprise (2010). I think Morton's argument is worthy of consideration, given that it too is concerned with developing a sophisticated argument about the relationship between American Evangelicalism and its entanglements with neoliberal capitalism.  

  1. Beth Morton, To Serve God and Wal-Mart (Reviewer 1, final point)

I am not familiar with this book, but I will read it shortly. Is it a necessary addition to the bibliography of this manuscript?

Reviewer 2 Report

This is an interesting and well-argued paper that deftly interweaves a rather personal narrative, concerning the author's parents involvement in Amway, with an incisive structural analysis of neoliberalism-as-religion. The interweaving of these two dynamics gives the paper moral and political weight, and also makes it more readable. 

The paper is thoughtful, solidly argued, and engaging. It makes a strong and interesting case, and it does so in a clearly structured way. I really enjoyed reading the paper and was struck by the rich terrain covered through a sharp focus on one particular instance of 'mystical capitalism'. I was frequently struck by other comparative instances of neoliberal Christianity, involving striking instrumentalizing, responsibilizing, and profit-oriented reconfigurations to Christian theology and practice.

There are a few places, however, that I think the paper could be further improved. 

I found the introduction of the 'decolonial' frame in the conclusion a little disconcerting. I had not considered the paper as a contribution to scholarship on decolonizing religion. Neither the abstract nor the introduction had indicated this. As a consequence, the introduction of 'decolonial Christianity' in the conclusion seemed out of place. This can be easily fixed by foregrounding this dynamic much earlier in the paper, and clearly signposting the conclusion as the culmination of the argument. 

Two further features about the conclusion also caused me concern. The first was the affirmation that Christianity cannot play a role in "dismantling the colonial matrix it helped found". This is framed as "a fair perspective" that ought to be accepted as valid. But the author clearly disagrees with this view, argument just a few paragraphs later that "neoliberalism and Christianity are not one and the same" and that a decoupling of Christianity and neoliberalism is a necessary measure. This can't be held both ways; and a clearer concluding argument is important.

Further, the argument that Christianity is irredeemably neoliberal/colonial is clearly an attempt to delegitimize and silence all those voices associated with Christianity. This can only be seen as a colonizing secular manoeuvre to coercively impose hegemonic discursive limits to political participation. In so doing it erases, for example, most of the indigenous voices of the Pacific who live and engage in the politics of decolonial liberation via the paradox of Christianity as both colonizing and decolonizing. As a mode of modern purification, the silencing of indigenous Christian voices advocated here is ethically dubious at best.     

It should be possible to pose a challenge about the possibility of decolonial Christianity without overstating this.

Second, also in the conclusion, while the coupling of Christianity and neoliberalism in the US context has been extensive, it is not helpful to argue that "perhaps" for neoliberalists "Christianity and neoliberalism are one and the same." This again overstates the argument, by rendering the American context into a universal one. A qualification here about the American locations of this statement would be valuable. 

For example, it is clearly not the case that Buddhist practitioners at the Dhammakaya Temple in Thailand, as described by Rachelle Scott in her book Nirvana for Sale? Buddhism, Wealth and the Dhammakaya Temple in Contemporary Thailand (2009) imagine themselves as Christian. Nor is this the case for Muslim activists in Egypt as described by Mona Atia in her book Building a House in Heaven: Pious Neoliberalism and Islamic Charity in Egypt (2013). Of course, the striking continuities between the author's argument in this paper and those described by Scott and especially Atia are thoroughly revealing for the ways in which neoliberal logics have become insinuated within religious dynamics across a diverse array of traditions. 

Finally, as I read the article I was frequently struck by similarities to the argument detailed in Beth Morton's book To Serve God and Wal-Mart: The Making of Christian Free Enterprise (2010). I think Morton's argument is worthy of consideration, given that it too is concerned with developing a sophisticated argument about the relationship between American Evangelicalism and its entanglements with neoliberal capitalism.  

These are all minor changes. As should be clear, I think the paper is a compelling and important contribution to debates about neoliberal religion.

Author Response

Response to Reviewer 2

I am grateful for the comments of the reviewers––both the positive statements and the incisive, insightful criticisms. Thank you for helping to improve this manuscript. I have broken down the critical comments into bullet points below. If I have somehow missed something, it was unintentional and I am happy to address further issues. 

  1. The most significant issue for the coherence of the paper is the final and concluding section. Its heading refers to Decolonial  Christianity and begins by asking whether there is any hope for Christiainity. This is odd because it was not the presenting issue of the paper, which related to why Amway could so effectively indoctrinate people. The discussion about decolonising Christianity is not irrelevant to the paper, but it needs to be introduced in ways that make its connection to the paper's stated aim much clearer and coherent.

  1. Introduce decolonial Christianity earlier and in abstract (Both reviewers)

This is a clear deficiency that I will fix by adding a few sentences peppered throughout the paper and in the abstract.

  1. Deal with “fair argument” against decolonial Christianity (Both reviewers)

The ambiguity here arises from the criticism of Christianity by non-Christian and post-Christian indigenous scholars (particularly George “Tink” Tinker), as well as my own, albeit unstated, philosophical stance. Ultimately, I see the possibility of decolonial Christianity (particularly from American evangelicals) as a groping in the darkness––I am hopeful, and eager to put forth the effort in its pursuit, but it is an enormous and unending task. It’s “accomplishment” entails a permanent process of revolution and rethinking in the face of new circumstances and criticisms, and it can never erase the history of atrocities committed. Nonetheless, I recognize that this kind of paradoxical approach appears merely contradictory in an article like this. I will address the issue as briefly as possible (i.e., without the explanation above) in the manuscript. 

  1. I think the title deserves to be more interrogative than its present descriptive mode. As the first sentence of the Abstract indicates, there is a question driving this paper (and it is answered) and it warrants being flagged in title. See my suggestion in my annotation.
  1. Interrogative element in title (Review 2, point 1)

I agree with this statement, and I will use the suggestion to devise a better title. I was unsure of the title, so I’m particularly grateful for this input. 

  1. I have some concerns about the biographical elements in the paper. It is fine for the author to draw on his/her own experiences of growing up in the Amway/Evangelical culture, but he/she also  makes claims about his/her deceased grandfather (and a power struggle he was involved in, p.9 ), his father (his vulnerabilty to Amway, p. 9) and his mother (her anger p.2) all of which have borderline moral judgements attached to them. There is no acknowledgement that these details and claims are made with the father's and mother's approval. I recognize that it's borderline, but in my own academic context such information would require approaval from a Human  Research Ethics Committee. 
  1. Biographical elements – parents approval?

I’m unsure of how the Human Research Ethics Committee might work, but if it’s worthwhile to work through that process, I’m open to it. The information from my mother’s experiences came from conversations and a brief, fairly informal interview conducted when I was back home in June. The information from my father came from conversations over the last year or so. It would take minimal effort to ask those questions again in a formal, recorded context. However, there are really only a handful of sentences that deal with experiences that are not directly my own, and it’s reasonable to excise those without significantly altering the meaning or impact of those sections.